# Standard of Care and Promising New Agents for the Treatment of Mesenchymal Triple-Negative Breast Cancer

**DOI:** 10.3390/cancers13051080

**Published:** 2021-03-03

**Authors:** Silvia Mezi, Andrea Botticelli, Giulia Pomati, Bruna Cerbelli, Simone Scagnoli, Sasan Amirhassankhani, Giulia d’Amati, Paolo Marchetti

**Affiliations:** 1Department of Radiological, Oncological and Pathological Science, University of Rome “Sapienza”, 00185 Rome, Italy; silvia.mezi@uniroma1.it (S.M.); bruna.cerbelli@uniroma1.it (B.C.); giulia.damati@uniroma1.it (G.d.); 2Department of Clinical and Molecular Medicine, University of Rome “Sapienza”, 00185 Rome, Italy; andrea.botticelli@uniroma1.it (A.B.); paolo.marchetti@uniroma1.it (P.M.); 3Department of Molecular Medicine, University of Rome “Sapienza”, 00185 Rome, Italy; 4Department of Medical and Surgical Sciences and Translational Medicine, University of Rome “Sapienza”, 00185 Rome, Italy; simone.scagnoli@uniroma1.it; 5Department of Plastic Surgery, Guy’s & St Thomas’ NHS Foundation Trust, London SE1 7EH, UK; sasan.am@gmail.com

**Keywords:** triple negative, breast cancer, mesenchymal subtype, immunotherapy, target therapy

## Abstract

**Simple Summary:**

Mesenchymal triple negative breast cancer subtype expresses genes involved in proliferation, epithelial to mesenchymal transition, stromal interaction and cell motility. Moreover, this subgroup is characterized by an immunosuppressive microenvironment. This review focuses on the intracellular pathways involved in tumorigenesis and cancer progression, as well as in the immune evasion mechanisms. Furthermore, we provide an overview of current clinical trials investigating the efficacy and safety of different therapeutic molecules for this aggressive subtype of triple negative breast cancer. The challenge is to restore immunocompetence by overcoming the chemo and immune-resistance profile of mesenchymal triple negative breast cancer to achieve a lasting response to therapy.

**Abstract:**

The pathologic definition of triple negative breast cancer (TNBC) relies on the absence of expression of estrogen, progesterone and HER2 receptors. However, this BC subgroup is distinguished by a wide biological, molecular and clinical heterogeneity. Among the intrinsic TNBC subtypes, the mesenchymal type is defined by the expression of genes involved in the epithelial to mesenchymal transition, stromal interaction and cell motility. Moreover, it shows a high expression of genes involved in proliferation and an immune-suppressive microenvironment. Several molecular alterations along different pathways activated during carcinogenesis and tumor progression have been outlined and could be involved in immune evasion mechanisms. Furthermore, reverting epithelial to mesenchymal transition process could lead to the overcoming of immune-resistance. This paper reviews the current knowledge regarding the mesenchymal TNBC subtype and its response to conventional therapeutic strategies, as well as to some promising molecular target agents and immunotherapy. The final goal is a tailored combination of cytotoxic drugs, target agents and immunotherapy in order to restore immunocompetence in mesenchymal breast cancer patients.

## 1. Introduction

The management of breast cancer (BC), the most common tumor in women [1] has improved since its sub-classification based on the expression of estrogen (ER), progesterone receptors (PR) [2] and human epidermal growth factor receptor-2 (HER2) [3]. Four main molecular subtypes of BC have been identified so far, based on the analysis of its global gene expression: Luminal A, luminal B, HER2, basal-like, and the more recently identified claudin-low tumor subtype [4,5].

Triple negative breast cancer (TNBC) is defined by the lack of expression of ER, PR and HER2 and accounts for about 10–20% of BCs [6]. This definition of TNBC is, however, limiting and does not allow for understanding its heterogeneous clinical behavior. Based on gene expression profiling, TNBCs were originally divided by Lehmann into six different subtypes: Basal-like 1 (BL1), basal-like 2 (BL2), immunomodulatory (IM), mesenchymal-like (M), mesenchymal stem cell-like (MSL) and luminal androgen receptor (LAR). Both M and MSL subtypes express genes involved in the epithelial to mesenchymal transition (ETM), stromal interaction and cell motility. Moreover, the M subgroup displays a higher expression of genes involved in proliferation, while the MSL more frequently shows expression of genes associated with cell stemness [5,7]. Further studies revealed that the IM and MSL subtypes were strongly influenced by the contribution of transcripts from normal stroma and immune cells of the tumor microenvironment, respectively. Thus, the classification has been refined in the following 4 subtypes: BL1, BL2, M and LAR [7,8,9]. The complexity of the TNBC classification is also determined by the possibility of identification of more than one subgroup in some histological types: For example, metaplastic breast carcinoma could belong to either the BL2 or the M subtype of BCs. Predominantly, metaplastic tumor cells are pleomorphic and arranged in solid nests (Figure 1A). The epithelial cells shows a diffuse positivity for the mesenchymal marker vimentin on immunohistochemical staining (Figure 1B).

For the last two decades chemotherapy has been the only therapeutic option, in the absence of a druggable target. It played a central role in the treatment of TNBC in all disease settings, even though each TNBC subtype has a different response pattern to the available chemotherapy regimens [10]. Gene expression analysis may also influence chemotherapy treatment choices; a randomized phase III trial was performed comparing use of carboplatin vs. docetaxel in unselected advanced TNBC and in a priori specified biomarker defined sub-populations [11]. In the unselected TNBC patient population, carboplatin was not more active compared to docetaxel (the standard of care). Furthermore, there was no evidence that basal-like biomarkers could be predictive of higher response to platinum. Conversely, in patients with breast cancer gene (BRCA) mutation, likely to have targetable defects in DNA repair, treatment with carboplatin doubled the response rate. Finally, the response to docetaxel was significantly better than the one to carboplatin in patients with non-basal-like TNBCs. Unfortunately, such poor platinum response results were not further stratified into M and LAR subtypes, given that the first one is known to be chemoresistant while the latter being responsive to anti androgenic hormone treatment. Targeted therapies have recently expanded the options for treatment, offering the potential for a dramatic change in clinical practice. Such is the case with Poli ADP-ribose polymerase (PARP) inhibitors in BRCA-mutated breast cancer [12,13], serine/threonine protein kinase (AKT) inhibitors in phosphatidylinositol-3-OH kinase (PI3K)/AKT/Phosphatase and tensin homologue (PTEN)-altered TNBC [14,15], and trophoblast cell surface antigen-2 (Trop-2) directed antibody-drug conjugate (ADC) sacituzumab govitecan in heavily pre-treated TNBC [16].

The immune cell profiling highlights that each TNBC subtype showed a correlation to an immunomodulatory pattern, with the exception of the M subgroup in which no immune infiltrate was shown, and which is characterized by an immunosuppressive microenvironment [17]. The immune suppression can influence the response to treatment in all settings [18,19,20]. The cold status of the M phenotype is extremely important in light of the results achieved with immunotherapy in TNBCs. As a matter of fact, immunotherapy has recently shown a significant impact on progression free survival (PFS) in association with chemotherapy in inoperable or metastatic TNBC [21].

This review summarizes the biological and clinical knowledge we have acquired to date in regards to the TNBC M subgroup and the state of the art of clinical trials investigating the efficacy and safety of different molecules, either on their own or in combination, in the treatment of this aggressive and resistant subtype of breast cancer.

## 2. Epithelial to Mesenchymal Transition and Immunosuppression

EMT is the change from an epithelial phenotype to a mesenchymal one and is often reversible. During EMT cells lose their mechanism of adhesion, thus promoting migration and metastasis. It has a predominant role in immune-escape mechanisms and chemoresistance. EMT could directly modulate expression of Programmed Death-Ligand 1 (PD-L1) and recruitment of macrophages in tumor microenvironment. PD-L1 is a T-cell inhibitory receptor, expressed on tumor cells and antigen presenting cells (APC); it leads to T-cell anergy and/or apoptosis upon ligation to its receptor programmed death-1 (PD-1) on T-cells in BC as well as in other malignancies [22,23,24,25].

PD-L1 expression in tumor cells is regulated by several mechanisms: It is often associated with the presence of inflammatory mediators, such as Interferon gamma (IFN-γ, a cytokine critical to both innate and adaptive immunity, and functions as the primary activator of macrophages and in stimulates natural killer cells and neutrophils) and the loss of PTEN with consequent PI3K activation. However, EMT induction was found to be able to upregulate PD-L1 expression in Claudin-low breast cancer subtype. On the other hand, PD-L1 modulation contributes to the sustainment of EMT in breast cancer cells implying the existence of a bidirectional crosstalk [24].

One of the main promoters of EMT is Tumor Growth Factor Beta (TGF-β), a multifunctional cytochine produced in the tumor microenvironment by tumor, stromal and immune cells [26]. TGF-β inhibits the action of several cytochines, including IFN-γ, TNF-α and IL-2, negatively affecting the immune response. Alterations in the TGF beta pathway therefore play a central role in immunosuppression and chemoresistance [27,28]. TGF beta inhibitors were found to be able to revert EMT and to block the expansion of chemotherapy-resistance cancer stem-like cells in vivo in mammary ephitelial cells [29,30]. TNBCs are often enriched with TGF-β ligands, supporting the idea that this pathway may be a promising target to be used in combination with both immunotherapy and chemotherapy [31,32]. One phase II study combining galunisertib, an anti TGF-β inhibitor, and radiotherapy in metastatic BC was recently terminated for lack of patient accrual (NCT02538471). Galunisertib was also evaluated in combination with paclitaxel in metastatic BC in a phase I study, actually active but not recruiting (NCT02672475). In vivo experiments have shown that the combination of galunisertib with PD-L1 blockade resulted in improved tumor growth inhibition and complete regressions in colon cancer models [33]. Three clinical studies are currently evaluating the potential synergy achievable by combining inhibition of TGF-β and PD-1/PD-L1 pathways in breast cancer (Table 1). These trials are evaluating Bintrafusp alfa (M7824) a bifunctional fusion protein composed of the extracellular domain of the human TGF-β receptor II fused to the C-terminus of each heavy chain of an IgG1 antibody blocking PD-L1.

This agent can act both as a TGF-β “trap”and as an anti PDL1 agent, resulting, in preclinical studies, in an increased density of CD8+ tumor-infiltrating lymphocytes in murine models and reversion of EMT in human lung cancer cell, as shown in preclinical studies [34]. Moreover, this drug showed encouraging activity in a phase I trial in no small cell lung cancer (NSCLC) [35]. In one of the these ongoing trial (NCT04296942), bintrafusp alfa was combined with Brachyury-TRICOM, a Modified Vaccinia Ankara (MVA) vector-based vaccine expressing the transgenes for the transcription factor brachyury, which drives the EMT [36].

Although data are not yet available, the strong rationale behind this molecules sustains great expectations towards them. Clinical trials focused on the inhibition of this fundamental mediator of ETM are ongoing (Table 1).

EMT causes the switch of the immune infiltrate from a neutrophilic pattern to a macrophage one. Macrophages in the tumor infiltrate could turn into macrophages of the M2 line, involved in the immunosuppression [37].

Reversing EMT could thus lead to overcoming the immune-resistance. On the other hand, inhibition of EMT could also modify the balance of the neutrophil infiltrate inducing the production of suppressor cells of myeloid derived origin (MDSCs), involved in a resistance mechanism to anti-EMT treatment [38,39]. The analysis of the M subtypes demonstrated increased expression of signaling pathway molecules, which are prominent in the processes of ETM. TGFβ, extracellular matrix (ECM)-receptor interaction, anaplastic lymphoma kinase (ALK), Wnt/β-catenin, and mammalian target of rapamycin mTOR, Rac1/Rho), decrease in E-cadherin expression, ECM receptor interaction and cell differentiation pathways (Wnt pathway, ALK pathway, and TGFβ signalling) (Table 2).

Wnt signaling pathway, which is involved in cell proliferation and signal transduc-tion, regulates the EMT. The Wnt pathway causes an accumulation of β-catenin in cyto-plasm and a subsequent translocation into the nucleus, where it activates transcriptional factors involved in EMT. Pan-PI3K inhibition could induce activation of Wnt pathway. The combination of two inhibitors showed activity in vitro and in vivo against TNBC cells lines [40].

For this reason, a combination of Wnt inhibitors and PI3K inhibitors could have a rationale in TNBC treatment in future clinical trials. A phase I trial, testing a combination of Wnt inhibitor and PI3K inhibitor, is currently ongoing (NCT03243331) (Table 1).

Activation of Wnt pathway upregulates PD-L1 expression in human TNBC stem cells [41]. A phase I clinical trial is testing a combination of Wnt inhibitors with anti PD-1 (NCT01351103).

## 3. Pathway Activation and Possible Therapeutic Targets

Several intracellular pathways are involved in tumorigenesis and cancer progression, as well as in immune evasion mechanisms (Table 2). Many of these pathways have been studied and evaluated as possible targets for therapy (Figure 2).

### 3.1. The NOTCH Pathway

The NOCTH pathway, which is implicated in the proliferation and differentiation of mammary BC stem cells by means of the activation of a membrane receptor plays a key role in tumor initiation (mediated by accumulation of MDSCs) and in tumor progression [42] (Table 2). The NOTCH receptors are also correlated with biological aggressiveness of BC, resistance to treatment, local and distant relapse. Efforts have been made to investigate the role of each NOTCH receptor in BC, however there are few data available from clinical trials when compared to the amount of preclinical evidence of activity of some molecules. γ-secretase inhibitors (GSIs) represent a possible way to target NOTCH pathway. These molecules can prevent the cleavage of the active form of the NOTCH receptor, blocking the intracellular signal and promoting cell apoptosis. Gastrointestinal toxicity highlighted in phase I trials however prevented research progression for GSIs [43].

Another strategy to reduce NOTCH activation involves the use of monoclonal antibodies able to bind the receptor and to convert it into the inactive form. Recently, tarexutumab, an anti-NOTCH monoclonal antibody, showed promising results in a preclinical study setting in a xenograft BC cells colony [44]. At the time of writing, an anti-NOTCH3 conjugated antibody was under investigation in a phase I trial in patients with BC and other solid tumors [45]. Preliminary results suggest that this molecule could be safe and effective. Moreover, several phase I and II studies with pathway NOCTH inhibitors are ongoing (Table 1).

### 3.2. PI3K/mTOR Pathway

The M subtype frequently displays abnormal activation in PI3K/mTOR pathway (Table 2). The same genetic aberration can be identified in other TNBC subgroups, such as luminal androgen receptor and BL1 [46,47].

Based on this evidence, several studies evaluated new agents against the PI3K/AKT pathway such as the BELLE-4 trial, a phase II/III study evaluating buparlisib including 338 patients of which 25% had TNBC [48]. This small pan-class I phosphoinositide 3-kinase inhibitor (which can be administered orally) was given in combination with either paclitaxel or placebo in advanced BC. This study did not show an advantage in combining buparlisib and paclitaxel in terms of PFS, in contrast with the results reported in the BELLE-2 and BELLE-3 trials [49]. Moreover, this trial did also fail to show a benefit from the addition of buparlisib to paclitaxel also in patients with TNBC, although the study population was not selected on the basis of genetic mutation.

Furthermore, there are two ongoing open label phase II clinical trials regarding the use of buparlisib in monotherapy (NCT01790932, NCT01629615) (Table 1). The phase II LOTUS trial included 124 patients with metastatic or locally advanced BC. These patients were randomized in order to receive either paclitaxel combined with ipatasertib (an orally bioavailable inhibitor of the serine/threonine protein kinase AKT), or paclitaxel alone. PFS increased from 4.9 to 6.2 months in the paclitaxel plus ipatasertib group. No benefit was shown in patients with low PTENprotein expression at immunohistochemistry. In 42 patients with PIK3/AKT1/PTEN mutated tumors, PFS increased from 4.9 to 9 months [14]. However, there was a correlated genetic alteration in only 28.57% of patients with PTEN loss. Overall survival (OS) increased from 16.9 to 25.8 months in patients treated with ipatasertib, mainly in patients with PI3K pathway altered tumors, as recently shown at the European Society of Medical Oncology (ESMO) 2020 congress [50]. An ipatasertib phase III trial is currently ongoing (NCT03337724) [51] (Table 1). Conversely, the preliminary results from the phase III trial IPATunity showed that ipasertinib in combination with paclitaxel failed to improve outcomes in terms of PFS and of objective response rate (ORR) in PIK3/AKT1/PTEN-altered ER positive BC, missing the primary and secondary endpoints of the study [52]. At present, a phase III study, evaluating the combination of ipatasertib with the anti PD-L1 atezolizumab and paclitaxel in metastatic/recurrent TNBC, is ongoing (NCT04177108) [53]. In addition, a phase II trial investigated another AKT inhibitor, capivasertib, which is a small molecule available orally. This molecule showed a benefit in PFS (from 4.2 months to 5.9) and in OS (from 12.6 to 19.1) in first line metastatic TNBC in combination with paclitaxel [15]. In a PI3K/AKT1/PTEN altered tumor population, PFS benefit increased from 3.7 to 9 months. PI3K/AKT/PTEN alteration could become an important biomarker of activity. The phase III trial of capivasertib and paclitaxel in first line TNBC is ongoing (CAPItello-2909, NCT03997123). A phase II trial studied ipatasertib vs. placebo in combination with paclitaxel in TNBC neoadjuvant setting. The rate of pathological complete response (pCR) was increased, although not significantly in patients treated with ipatasertib, predominantly in PI3K/AKT/PTEN altered tumors and in tumors with PTEN loss (39% vs. 9% and 32% vs. 6%, respectively) [54].

New combinations of PI3K inhibitors with PARP inhibitors (PARPi) and AKT inhibitors with PARPi are being tested in an ongoing phase I and II trial (NCT0257644). A phase I study explored the combination of liposomal doxorubicin, bevacizumab and mTOR inhibitors (either temsirolimus or everolimus), including 52 patents with metaplastic breast cancer [55]. The selection of metaplastic histotype is relevant, considering the frequency of PI3K pathway alterations and the high expression of angiogenetic factors in this kind of disease, frequently associated to the M subgroup. The ORR in this study was 21% and the clinical benefit rate was 40%. PI3K pathway aberrations were associated with a significant improvement in the ORR (31% vs. 0%).

The same authors compared efficacy of mTOR based chemotherapy in metaplastic and non- metaplastic TNBC. In the first setting, mTOR-based therapy was significantly associated with better PFS and OS [56].

An ongoing phase I study is testing the combination of everolimus and eribulin in pretreated metastatic TNBC [NCT02616848]. As a matter of fact, in the Embrance phase III trial, eribulin was associated with a significant increase in OS in heavily pretreated metastatic BC and was more effective in TNBC [13]. The main mechanism of resistance to mTOR inhibitors is due to the rebound of AKT activity. Therefore, a combination of mTOR inhibitors and AKT inhibitors could have a strong rationale, due to AKT activation following the PI3K inhibition. Single PI3K pathway agent-based therapy frequently leads to development of resistance [57].

Several ongoing trials are testing drugs targeting the PI3K pathway in TNBC in combination with other agents. It will be important to focus clinical trials towards PIK3CA/AKT1/PTEN mutated tumors in future. Moreover, PTEN loss seems to correlate with immune-resistance, as evidenced in a preclinical study on a melanoma model [58]. PI3K/mTOR pathway contributes to MDSC accumulation promoting Granulocyte Colony-Stimulating Factor (G-CSF) production [46]. Consequently, ongoing trials are testing combination strategy with immunotherapy in TNBC (NCT03424005 and NCT03395899).

### 3.3. The Molecular Chaperone HSP90

Heat Shock Proteins90 (HSP90) HSP90 is a highly conserved molecular chaperone interacting with several proteins regulating cell proliferation, including receptor tyrosine kinases and mesenchymal-epithelial transition factor (MET), transcription factors, such as Hypoxia-Inducible Factor (HIF)-1 amd Tumor protein (TP) 53, signaling proteins such as AKT and Rous sarcoma oncogene cellular homolog (Src) and cell cycle regulatory proteins (Cicline dependent kinase (CDK) 4 and 6). The inhibition of HSP90 could allow to disrupt multiple oncogenic pathways simultaneously [59]. In a phase II clinical trial, ganetespib showed little evidence of activity in TNBC with an acceptable toxicity profile [60,61]. Several clinical trials are ongoing (Table 1).

One possible resistance mechanism to HSP90 inhibitors could involve the upregulation of Janus-family tyrosine kinase (JAK)/Signal transducer and activator of transcription (STAT) signaling pathways [62]. STAT3 plays an important role in cancer growth, metastasis, resistance to treatment (including chemotherapy and other target therapies) and immune evasion in TNBC. In vivo and in vitro preclinical studies showed the ability of STAT3 inhibitors to control the growth of BC cells [63] (Table 1).

### 3.4. The Src Family Kinases

Src is a non-receptor tyrosine kinase, member of the Src family kinases (SFKs), whose activity is related to cellular proliferation, differentiation, survival, migration and angiogenesis in both normal and cancer cells (Table 2) [64].

Src is composed of several domains: A carboxy-terminal region, a unique amino-terminal domain and four Src homology (SH) domains that are responsible for the interaction with activated Receptor tyrosine kinases (RTKs) at the plasma membrane, including IGF-1R, EGFR, HER2, Platelet-Derived Growth Factor (PDGF) and cMET [65]. These interactions lead to the activation of signaling pathways involved in cellular growth, such as RAS-MAPK, PI3K/AKT and STAT (pathways). Besides, Src in conjunction with focal adhesion kinase (FAK) acts as regulator of integrin-dependent attachment, focal adhesion turnover and cell migration [66]. It plays a central role in tumor invasion and motility and its deficiency results in defects in cytoskeletal organization and spreading.

Src may also promote loss of epithelial adhesion and cell scattering through modulation of cell–cell adhesion [67]. Src protein levels and Src protein kinase activity have been observed to be frequently increased in human neoplastic tissue and in human cancer cell lines, however this corresponds to Src gene mutation or amplification only in few cases [68]. Src is more often activated by receptors tyrosin-kinases receptors such as EGFR.

There is a synergistic relationship between EGFR and Src, which is both activated by and activates the receptor. Src is responsible for the phosphorylation of tyrosin residues Y845 and Y1101 on EGFR, which leads to an enhancement in EGF-mediated DNA synthesis [69]. Overexpression of Src increases the response of EGFR-mediated processes by integrating EGFR with other non-related membrane receptors and intracellular signaling molecules [69,70]. Src activation has been described as a determinant of resistance to anti-EGFR drugs in human lung, colorectal and pancreatic cancer cell models [71]. For example, Src contributes to c-MET activation in the EGFR-TKI gefitinib resistant non-small cell lung cancer cells [72]. Moreover, in BC cells, MET and src were found to cooperate to overcome gefitinib induced EGFR inhibition [73].

Therefore, Src is a key substrate in the transduction pathways, which are mediated by both EGFR and IGF-IR, as demonstrated by the inhibition of migration, which is induced by EGFR/IGF-IR via a Src inhibitor in claudin-low cell lines.

Src kinase family representatives were a promising target in TNBC. Overexpression of Src kinase in TNBC led to the investigation of dasatinib, an oral, small molecule multi-kinase inhibitor (BCR/ABL and Src family tyrosine kinase inhibitor) in preclinical and clinical studies. Unfortunately, clinical studies showed a rapid development of resistance [74,75,76]. Despite extensive preclinical evidence, which warrants targeting src as a promising therapeutic approach for cancer, Src inhibitors showed only a minimal therapeutic activity in several types of solid tumors when used as a single agent. The future of targeting Src as a cancer therapy appears gloomy if a different approach combining specific inhibitors in selected patients will not be envisioned. A phase II study of dasatinib and paclitaxel in metastatic BC, including 40 patients, was stopped early due to slow accrual. Dasatinib had modest activity with an ORR of 23%, but 80% of the patients had ER positive BC [77].

### 3.5. EGFR Overexpression

Several types of cancer, including BC, involve deregulation of EGFR-mediated signaling caused by different molecular mechanisms, such as overexpression, acquisition of activating mutations of the receptor and activation induced by ligands, which act in autocrine/paracrine manner [78]. The receptor regulates many aspects of the tumor behavior including cell proliferation, migration, angiogenesis and is involved in development and progression of BC. EGFR is frequently overexpressed in TNBC. BL2 TNBC subtype displays unique gene ontologies involving growth factors signaling. EGFR has also been implicated as a key role player in the mitogenic and motogenic effects. Recent studies have shown that EGFR regulate migration, tumor invasion and EMT. EGFR inhibitors induced a restoring from mesenchymal to epithelial phenotype in TNBC cells and the EGFR TKIs erlotinib inhibited tumor growth and metastasis in a SUM149 xenograft mouse model, showing an antimetastatic effect that could be the basis of “overlap sensitivity” to dasatinib between M and BL2 subtypes [79].

In the M subtype, EGFR is correlates to MAP kinase and PI3K pathway and to the downstream Src pathways upregulation. Several preclinical and clinical studies have evaluated the efficacy of EGFR inhibitors in TNBC. The anti-EGFR monoclonal antibody cetuximab in combination with cisplatin or carboplatin showed modest activity in metastatic setting in less than 20% of patients. EGFR pathway probably has at least one alternative mechanism of activation in TNBC [80,81]. In addition, in a phase II trial, cetuximab and panitumumab showed modest activity in a neo-adjuvant setting in association with antracyline- and taxane-based chemotherapy [82]. Panitumumab in association with chemotherapy had 46.8% of pCR rate, while cetuximab in combination with chemotherapy had 24% of pCR rate [83]. A phase II study showed a RR of 18% in TNBC vs. 0% of cetuximab in combination with irinotecan in non-TNBCs [84]. These clinical studies showed that targeting EGFR in BC yielded no credible results. It is now clear that EGFR inhibition alone is unlikely to provide disease control in most TNBCs; combination strategies targeting other components of the pathway and dedicated tissue-based studies are likely to be necessary. Despite the fact that EGF and Src have strong mitogenic and pro-migratory properties and promote metastasis, so far, the strategy of targeting just one of them seems not to be enough to inhibit tumor behavior. Therefore, the eventual clinical remission of patients who are not selected on the basis of predictors of response will be only transient, because of the development of drug resistance. Other studies testing cetuximab, panitumumab and erlotinib are currently ongoing (NCT03692689, NCT02593175, NCT02876107) (Table 1).

## 4. Angiogenesis

Agents targeting neo-angiogenesis have still a predominant role in TNBCs, which feature high levels of vascular endothelial growth factors (VEGF). Bevacizumab, an anti-VEGF monoclonal antibody, is the most widely studied antiangiogenetic drug in many solid tumors. Bevacizumab has confirmed its activity mainly in TNBC. Metaplastic BC, similar to cancer stem cells-derived tumors, often expresses a high level of VEGF and hypoxia inducible factor (HIF). In vitro studies showed that the specific inhibitor of mTOR temsirolimus is able to decrease the levels of VEGF and HIF, supporting a possible synergistic relationship with Bevacizumab [85,86,87]. Bevacizumab is currently still recommended by the European Medicines Agency (EMA) in first line metastatic HER2 negative BC, although no OS advantage was demonstrated [88,89].

VEGF-2 is a significant negative prognostic biomarker in TNBC [90]. Similarly, in The RIBBON 1 trial, bevacizumab combined with different chemotherapy regimens significantly improved PFS (but not OS) in first line metastatic HER2 negative BC [91]. The Phase III RIBBON-2 study furthermore showed that bevacizumab, in combination with different therapeutic options achieved an advantage in both PFS and ORR in second line HER2 negative BC, although OS was not significantly improved [92,93]. However, the clinical trial population was heterogeneous and not analyzed for molecular subtypes. In a randomized phase III trial baseline pVEGF-A level was not useful in identifying patients who would benefit the most from bevacizumab [89].

A phase II trial recently studied the usage of nab-paclitaxel and bevacizumab followed by erlotinib and bevacizumab in a maintenance strategy; 74% of unselected metastatic TNBC patients experienced a partial response, but no significant results in PFS were achieved [94].

In a neoadjuvant setting, bevacizumab in combination with doxorubicin, cyclophosphamide and paclitaxel showed efficacy and safety with a 42% of pCR in TNBC, without tumor subtype selection. Moreover, in a phase II trial, pCR was reported in 15% of patients treated with 4 cycles of cisplatin in combination with 3 cycles of bevacizumab, however toxicity limited the completion of neoadjuvant therapy in 11% of patients [95].

The phase III trial BEATRICE assessed the role of bevacizumab in an adjuvant setting, but in the preliminary results, showed no benefit in terms of OS was detected in the bevacizumab arm compared to the chemotherapy only one [96]. Several ongoing clinical trials are testing angiogenesis inhibiting molecules, either alone or in combination in an effort to improve outcomes in TNBC and in M sub-type (Table 1).

## 5. Immune System and Immunotherapy

Immunotherapy is promising in treating TNBCs. However, not all TNBCs are equally susceptible, nor do they have the same immunological features. As previously mentioned, in the M subtype, the immune-escape mechanism could be predominant (Table 2). Consequently, in this subgroup, the best strategy could be to revert a non-responsive tumor to a responsive one through the combination of immunotherapy with other agents. A combination of atezolizumab with nab-paclitaxel prolonged PFS in a recent phase III trial [97]. In the treatment group, median PFS was 7.2 months (95% CI 5.6–7.4) with atezolizumab and 5.5 months (5.3–5.6) with placebo (HR 0.80 [95% CI 0.69–0.92, *p* = 0.0021]). In the PD-L1 positive population, the median PFS was significantly longer in the atezolizumab group (7.5 months) than in the placebo group (5.3 months, HR 0.63, *p* < 0.0001). No difference in PFS was reported in the PD-L1 negative subgroup. The final exploratory analysis showed an increased OS in PD-L1 positive subgroup (25 vs. 18 months, HR 0.71). Median OS was 21.0 months with atezolizumab and 18.7 months with placebo (HR 0.86, *p* = 0.078) [98].

Data from the phase Ib KEYNOTE 173 trial suggest that the association of the anti PD-1 pembrolizumab and different chemotherapy schedules is effective in neoadjuvant setting in locally advanced TNBCs, with an overall pCR of 60%, an ORR from 70% to 100% and a manageable safety profile [99]. In a recent phase III trial, the addition of pembrolizumab to the standard neoadjuvant chemotherapy increased the pCR rate from 51.2% in the placebo arm to 64.8% in the combination arm (*p* < 0.001). Moreover, after a median follow-up of 15.5 months, 7.4% in the pembrolizumab group and 11.8% in the placebo group had local or distant recurrence or passed away (HR 0.63; 95% CI, 0.43 to 0.93). The efficacy of pembrolizumab in combination with chemotherapy in metastatic TNBC was also evaluated in the phase 3 trial KEYNOTE 355. In this study, patients were stratified according to PD-L1 status (combined positive score [CPS] < 1 or ≥1), previous treatment received, chemotherapy backbone (carboplatin plus gemcitabine or taxanes) but not according to the molecular subtype of TNBC. In patients with CPS of 10 or more, median PFS was 9.7 months in the treatment group and 5.6 months in the control one (HR 0.65, *p* = 0.0012). PFS rate at 12 months was significantly higher in the pembrolizumab group than in the placebo group (39.1% vs. 23.0%). There was no significant difference in PFS between treatments in patients with CPS ≥ 1 (7.6 vs. 5.6 months, HR 0.74, one-sided *p* = 0.0014); prespecified statistical criterion of alpha = 0.00111. However, the rate of PFS at 12 months in patients with CPS ≥ 1 was higher in the combination group than in the chemotherapy group (31.7% vs. 19.4%). No difference in PFS was achieved in patients with PD-L1 CPS < 1 (median PFS 6.3 months in the pembrolizumab–chemotherapy group and 6.2 months in the placebo–chemotherapy group; HR 1.08) [100]. Finally, the efficacy of durvalumab, another anti PD-L1 agent, was evaluated in neoadjuvant setting both in combination with chemotherapy or alone in a window-phase pre neoadjuvant chemotherapy (NACT). No statistical difference in pCR rate was seen between durvalumab group and placebo group (53.4% vs. 44.2%, OR = 1.45, unadjusted Wald *p* = 0.224). However, pCR rate was significantly higher in the window-phase group who received durvalumab alone before starting chemotherapy (61.0% vs. 41.4%, OR = 2.22, 95%, *p* = 0.035) [101].

These results highlight the role of immunotherapy in both metastatic and neoadjuvant setting as an additional strategy in TNBC and the role of PD-L1 as a predictive biomarker. However anti PD-1 agents, either alone or in combination with chemotherapy, seem to be ineffective in PD-L1 negative patients, as a result of a primary resistance that could be at least partly explained by the presence of mesenchymal transition. Furthermore, even in highly selected PD-L1 positive population, about 70% of patients will experience a progression of disease in the first 12 months. Further in-depth genomic investigations are required to understand and overcome resistance mechanisms. As an example, preliminary results from the IMpassion 131 trial are in contrast to those of previous studies, with no benefit in clinical outcomes from a combination of atezolizumab plus weekly paclitaxel vs. weekly paclitaxel alone [102]. PFS was not significantly improved by the combination of drugs vs. chemotherapy alone in either the PD-L1–positive (6.0 vs. 5.7 months; HR = 0.82; *p* = 0.20) or in the intention-to-treat population (5.7 vs. 5.6 months; HR 0.86; significance not formally tested). No subgroup had an additional benefit from the addition of anti PD-L1. Moreover, the combination did not improve OS in the PD-L1–positive group (22.1 vs. 28.3 months; HR 1.12).

A different distribution of molecular subtypes in the population of IMpassion 130 and 131 trials could be a reason for these results, together with the use of a higher dose of corticosteroids or the different immunomodulation of taxanes. Different combinations of anti PD-L1 agents and chemotherapy are currently being evaluated. Preliminary results from the ENHANCE phase I study showed that the combination of eribulin and pembrolizumab in metastatic TNBC resulted in an ORR in the PD-L1 positive and PD-L1 negative subgroups of 34.5% and 16.1%, respectively. Complete response was achieved in three cases, one of which was in a patient with a PD-L1–negative tumor. The combination was active and effective regardless of PD-L1 status or prior treatment with chemotherapy. Patients who received a combination as a first line had a response rate of 29.2 % vs. 22% in patients who received one or two prior lines [103]. Several clinical trials are ongoing in order to improve outcomes in TNBC and in M sub-type (Table 1).

Within these studies, some are evaluating immunotherapy in combination strategy for treatment of TNBC, unfortunately without any distinction in molecular subgroups [104].

## 6. Future Perspectives

TNBC is an extremely heterogeneous entity in terms of genetic and molecular characteristics. This heterogeneity is reflected in a different clinical behavior in terms of prognosis and response to the therapies available today and requires the identification of molecular and genetic factors able to drive customized therapeutic choices. Lehmann’s classification allows to identify different subgroups of TNBC, sharing the same immunohistochemical definition but distinct in genetic and molecular alterations involved in carcinogenesis, tumor growth and metastasis. Each TNBC may have biological features partly common to one or more of Lehmann’s subgroups. How can we translate our genetic and molecular knowledge into daily clinical practice? It seems difficult to give a precise prognostic and predictive connotation to the TNBC M subtype, given the multiple possibilities of interaction between the different pathways regulating cell growth and death (Figure 2).

Further effort will be needed in future in order to define the rationale and purpose of clinical trials and design specific clinical trials for the different molecular subtypes exploiting the characteristics of each subtype. The immunohistochemical definition alone cannot guarantee a correct selection of patient population in clinical trials; the study design must include a genetic and molecular profiling of tumors, identifying a more homogeneous population in terms of molecular features. This is a complex task, given the infinite possible interactions between hormonal receptors, DNA repair systems, cell cycle regulation systems, angiogenesis and immune system. In this scenario, other promising targets such as the Activating Transcription Factor 4 (ATF4) may represent a valuable prognostic biomarker and therapeutic target in patients with TNBC, since it is able to modulate TGFβ-induced aggression in TNBC via SMAD2/3/4 and mTORC2 signaling. ATF4 is overexpressed in TNBC patients, and in vitro studies have shown increased levels of ATF4 in TGFβ1 treated TNBC cell lines. ATF4 is involved in the regulation of signaling pathways associated with tumor metastases, proliferation and drug resistance. Furthermore, inhibition of ATF4 expression led to a reduction in migration, invasiveness, proliferation, ETM and levels of antiapoptotic and stem cell markers and correlated with lower patient survival [105]. The main objective so as to modulate the aggressiveness of will be to identify integrated therapeutic strategies adapted to the genetic tumor identity. Patient profiling will therefore be critical in order to determine both the first and the subsequent lines of combination treatment. Monitoring tumor signaling molecules levels during treatment will be required in order to achieve a durable response. The final goal will be a tailored combination of cytotoxic agents, target agents and immunotherapy with the main challenge of restoring immunocompetence in M breast cancer patients.

## 7. Conclusions

Mesenchymal TNBC subtype is characterized by the expression of genes involved in ETM, stromal interaction, cell motility, proliferation and by immune evasion.

ETM plays a predominant role in chemoresistance and immune-escape, by means of both upregulation of the expression of PD-L1 and recruitment of macrophages in tumor microenvironment. EGFR and Src are key role players in the mitogenic and motogenic effects in M subtype as well and regulate many aspects of tumor behavior including cell proliferation, migration and angiogenesis. Moreover, several molecular alterations along different pathways which are activated during carcinogenesis and tumor progression could be involved in M subtype immune evasion and resistance. The M subtype frequently displays abnormal activation in the PI3K/mTOR pathway and PTEN loss, which contributes to immune-resistance by MDSC accumulation resulting in an immunosuppressive microenvironment. These complex and interconnected alterations lead to a mobile and infiltrating tumor phenotype able to escape immune control. Reverting the ETM process could lead to the overcoming of immune-resistance, inhibiting the proliferation of chemo-resistant cells and reverting a non-responsive tumor to a responsive one.

Nevertheless, these interconnected pathways, showing cross-talk and transactivation between their different components demonstrate the need for combined blockade strategies in order to obtain a long-lasting control of the disease.

The final goal will be a tailored combination of cytotoxic agents, target agents and immunotherapy so as to obtain a reduction in migration, invasion and proliferation, restoring immunocompetence in M breast cancer patients to improve not only disease control but also, finally, patient survival.

## Figures and Tables

**Figure 1 cancers-13-01080-f001:**
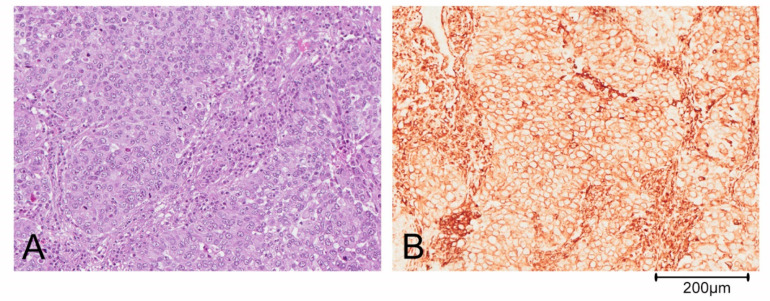
Metaplastic breast carcinoma. (**A**) Tumor cells are highly pleomorphic and arranged in solid nests. (**B**) Immunohistochemistry shows a diffuse positivity of neoplastic cells for the mesenchymal marker vimentin.

**Figure 2 cancers-13-01080-f002:**
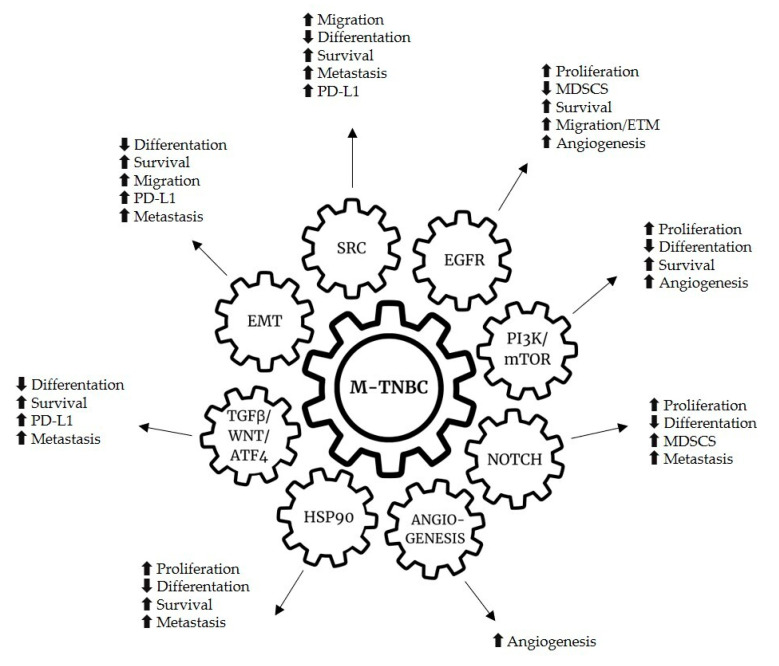
Genetic alterations and hallmarks of mesenchymal breast cancer triple negative subtype.

**Table 1 cancers-13-01080-t001:** Comprehensive overview of currently ongoing clinical trials evaluating the pathways expressed in triple negative mesenchymal breast cancer as targets for therapy.

Target	NCT Number	Title	Status	Interventions	Phase
NOTCH	NCT04461600	A Study of AL101 Monotherapy in Patients With Notch Activated Triple Negative Breast Cancer	Recruiting	Drug: AL101	2
	NCT01238133	Gamma-Secretase/Notch Signalling Pathway Inhibitor RO4929097, Paclitaxel, and Carboplatin Before Surgery in Treating Patients With Stage II or Stage III Triple- Negative Breast Cancer	Terminated	Drug: Carboplatin Drug: Gamma-Secretase Inhibitor RO4929097 Other: Laboratory Biomarker Analysis Drug: Paclitaxel Other: Pharmacological Study Procedure: Therapeutic Conventional Surgery	1
	NCT01071564	RO4929097 and Vismodegib in Treating Patients With Breast Cancer That is Metastatic or Cannot Be Removed By Surgery	Terminated	Drug: Gamma-Secretase Inhibitor RO4929097 Other: Laboratory Biomarker Analysis Other: Pharmacogenomic Study Other: Pharmacological Study Drug: Vismodegib	1
PIK3CA—PTEN—MTOR	NCT04251533	Study Assessing the Efficacy and Safety of Alpelisib + Nab-paclitaxel in Subjects With Advanced TNBC Who Carry Either a PIK3CA Mutation or Have PTEN Loss Without PIK3CA Mutation	Recruiting	Drug: alpelisib Drug: placebo Drug: nab-paclitaxel	3
	NCT04216472	Nab-paclitaxel and Alpelisib for the Treatment of Anthracycline Refractory Triple Negative Breast Cancer With PIK3CA or PTEN Alterations	Recruiting	Drug: Alpelisib Drug: Nab-paclitaxel	2
	NCT03337724	A Study of Ipatasertib in Combination With Paclitaxel as a Treatment for Participants With PIK3CA/AKT1/PTEN- Altered, Locally Advanced or Metastatic, Triple-Negative Breast Cancer or Hormone Receptor-Positive, HER2- Negative Breast Cancer	Active, not recruiting	Drug: Ipatasertib Drug: Paclitaxel Drug: Placebo	3
	NCT01623349	Phase I Study of the Oral PI3kinase Inhibitor BKM120 or BYL719 and the Oral PARP Inhibitor Olaparib in Patients With Recurrent Triple Negative Breast Cancer or High Grade Serous Ovarian Cancer	Active, not recruiting	Drug: BKM120 and Olaparib Drug: BYL719 and Olaparib	1
	NCT01920061	A Study Of PF-05212384 In Combination With Other Anti-Tumor Agents and in Combination With Cisplatin in Patients With Triple Negative Breast Cancer in an Expansion Arm (TNBC)	Completed	Drug: PF-05212384 (gedatolisib) Drug: Docetaxel Drug: Cisplatin Drug: Dacomitinib	1
	NCT03961698	Evaluation of IPI-549 Combined With Front-line Treatments in Pts. With Triple-Negative Breast Cancer or Renal Cell Carcinoma (MARIO-3)	Recruiting	Drug: IPI-549 (eganelisib) Drug: Atezolizumab Drug: nab-paclitaxel Drug: Bevacizumab	2
	NCT03853707	Ipatasertib in Combination With Carboplatin, Carboplatin/Paclitaxel, or Capecitabine/Atezolizumab in Treating Patients With Metastatic Triple Negative Breast Cancer	Recruiting	Drug: Atezolizumab Drug: Capecitabine Drug: Carboplatin Drug: Ipatasertib Drug: Paclitaxel Other: Quality-of-Life Assessment Other: Questionnaire Administration	1-2
	NCT03243331	An Initial Safety Study of Gedatolisib Plus PTK7-ADC for Metastatic Triple-negative Breast Cancer	Recruiting	Drug: Gedatolisib Drug: PTK7-ADC	1
	NCT04177108	A Study Of Ipatasertib in Combination With Atezolizumab and Paclitaxel as a Treatment for Participants With Locally Advanced or Metastatic Triple- Negative Breast Cancer.	Recruiting	Drug: Atezolizumab Drug: Ipatasertib Drug: Paclitaxel Drug: Placebo for Atezolizumab Drug: Placebo for Ipatasertib	3
	NCT03805399	FUSCC Refractory TNBC Umbrella (FUTURE)	Recruiting	Drug: Pyrotinib with Capecitabine Drug: AR inhibitor with CDK4/6 inhibitor Drug: anti PD-1 with nab-paclitaxel Drug: PARP inhibitor included therapy Drug: BLIS with anti-VEGFR included therapy Drug: MES with anti-VEGFR included therapy Drug: mTOR inhibitor with nab- paclitaxel	1-2
	NCT02583542	A Study of AZD2014 in Combination With Selumetinib in Patients With Advanced Cancers	Active, not recruiting	Drug: AZD2014 Drug: AZD6244	1-2
	NCT02531932	Comparison of Single-Agent Carboplatin vs. the Combination of Carboplatin and Everolimus for the Treatment of Advanced Triple-Negative Breast Cancer	Recruiting	Drug: Carboplatin Drug: Everolimus	2
	NCT02890069	A Study of PDR001 in Combination With LCL161, Everolimus or Panobinostat	Recruiting	Biological: PDR001 Drug: LCL161 Drug: Everolimus Drug: Panobinostat Drug: QBM076 Drug: HDM201	1
	NCT02456857	Liposomal Doxorubicin, Bevacizumab, and Everolimus in Patients With Locally Advanced TNBC With Tumors Predicted Insensitive to Standard Chemotherapy; A Moonshot Initiative	Recruiting	Biological: Bevacizumab Drug: Everolimus Other: Laboratory Biomarker Analysis Drug: Pegylated Liposomal Doxorubicin Hydrochloride	2
	NCT02120469	Eribulin Mesylate and Everolimus in Treating Patients With Triple-Negative Metastatic Breast Cancer	Active, not recruiting	Drug: everolimus Drug: eribulin mesylate Other: pharmacological study Other: laboratory biomarker analysis	1
TGF beta	NCT03579472	M7824 and Eribulin Mesylate in Treating Patients With Metastatic Triple Negative Breast Cancer	Recruiting	Drug: Anti-PD-L1/TGFbetaRII Fusion Protein M7824 Drug: Eribulin Mesylate	1
	NCT04296942	BN-Brachyury, Entinostat, Adotrastuzumab Emtansine and M7824 in Advanced Stage Breast Cancer (BrEAsT)	Recruiting	Biological: Brachyury-TRICOM Drug: Entinostat Biological: M7824 Biological: Ado-trastuzumab emtansine	1
	NCT04489940	Bintrafusp Alfa in High Mobility Group AT-Hook 2 (HMGA2) Expressing Triple Negative Breast Cancer	Recruiting	Drug: Bintrafusp alfa	2
	NCT02672475	Galunisertib and Paclitaxel in Treating Patients With Metastatic Androgen Receptor Negative (AR-) Triple Negative Breast Cancer	Active, not recruiting	Drug: Galunisertib Other: Laboratory Biomarker Analysis Drug: Paclitaxel	1
WNT	NCT01351103	A Study of LGK974 in Patients With Malignancies Dependent on Wnt Ligands	Recruiting	Drug: LGK974 Biological: PDR001	1
HSP90	NCT02474173	Onalespib and Paclitaxel in Treating Patients With Advanced Triple Negative Breast Cancer	Active, not recruiting	Other: Laboratory Biomarker Analysis Drug: Onalespib Drug: Paclitaxel Other: Pharmacological Study	1
	NCT02898207	Olaparib and Onalespib in Treating Patients With Solid Tumors That Are Metastatic or Cannot Be Removed by Surgery or Recurrent Ovarian, Fallopian Tube, Primary Peritoneal, or Triple- Negative Breast Cancer	Active, not recruiting	Drug: Olaparib Drug: Onalespib	1
	NCT03654547	Safety of TT-00420 Monotherapy in Patients With Advanced Solid Tumors and Triple Negative Breast Cancer	Recruiting	Drug: TT-00420	1
JAK-STAT	NCT02876302	Study Of Ruxolitinib (INCB018424) With Preoperative Chemotherapy For Triple Negative Inflammatory Breast Cancer	Recruiting	Drug: Ruxolitinib Drug: Paclitaxel Drug: Doxorubicin Drug: Cyclophosphamide	2
EGFR	NCT02720185	Window of Opportunity Trial of Dasatinib in Operable Triple Negative Breast Cancers With nEGFR	Recruiting	Drug: Dasatinib Procedure: Conventional Surgery Other: Laboratory Biomarker Analysis	2
	NCT04603287	A Study of SI-B001, an EGFR/HER3 Bispecific Antibody, in Locally Advanced or Metastatic Epithelial Tumors	Recruiting	Drug: SI-B001	2
	NCT02876107	Carboplatin and Paclitaxel With or Without Panitumumab in Treating Patients With Invasive Triple Negative Breast Cancer	Recruiting	Drug: Carboplatin Other: Laboratory Biomarker Analysis Drug: Paclitaxel Biological: Panitumumab	1
	NCT04429542	Study of Safety and Tolerability of BCA101 Alone and in Combination With Pembrolizumab in Patients With EGFR-driven Advanced Solid Tumors	Recruiting	Drug: BCA101 Drug: Pembrolizumab	1
	NCT02593175	Women’s MoonShot: Neoadjuvant Treatment With PaCT for Patients With Locally Advanced TNBC	Recruiting	Drug: Carboplatin Other: Laboratory Biomarker Analysis Drug: Paclitaxel Biological: Panitumumab	2
AKT	NCT01520389	Safety Study of the Drug MM-151 in Patients With Advanced Solid Tumors Resisting Ordinary Treatment	Completed	Drug: MM-151 Drug: MM-151 + irinotecan	1
CDK	NCT04553133	PF-07104091 as a Single Agent and in Combination Therapy	Recruiting	Drug: PF-07104091 monotherapy Drug: PF-07104091 + palbociclib Drug: PF-07104091 + palbociclib + letrozole	2
	NCT03519178	A Safety, Pharmacokinetic, Pharmacodynamic and Anti-Tumor Study of PF-06873600 as a Single Agent and in Combination With Endocrine Therapy	Recruiting	Drug: PF-06873600 Drug: Endocrine Therapy 1 Drug: Endocrine Therapy 2	2
ANGIOGENESIS	NCT03170960	Study of Cabozantinib in Combination With Atezolizumab to Subjects With Locally Advanced or Metastatic Solid Tumors	Recruiting	Drug: cabozantinib Drug: atezolizumab	1-2
	NCT02187991	Study to Compare Alisertib With Paclitaxel vs. Paclitaxel Alone in Metastatic or Locally Recurrent Breast Cancer	Active, not recruiting	Drug: Paclitaxel Drug: Alisertib	2
	NCT03577743	Effect of Bevacizumab in Metastatic Triple Negative Breast Cancer	Recruiting	Drug: Bevacizumab	2
	NCT04408118	First Line Atezolizumab, Paclitaxel, and Bevacizumab (Avastin^®^) in mTNBC	Recruiting	Drug: Atezolizumab Drug: Paclitaxel Drug: Bevacizumab	2
	NCT03961698	Evaluation of IPI-549 Combined With Front-line Treatments in Pts. With Triple-Negative Breast Cancer or Renal Cell Carcinoma (MARIO-3)	Recruiting	Drug: IPI-549 (eganelisib) Drug: Atezolizumab Drug: nab-paclitaxel Drug: Bevacizumab	2
	NCT03424005	A Study Evaluating the Efficacy and Safety of Multiple Immunotherapy-Based Treatment Combinations in Patients With Metastatic or Inoperable Locally Advanced Triple-Negative Breast Cancer	Recruiting	Drug: Capecitabine Drug: Atezolizumab Drug: Ipatasertib Drug: SGN-LIV1A Drug: Bevacizumab Drug: Chemotherapy (Gemcitabine + Carboplatin or Eribulin) Drug: Selicrelumab Drug: Tocilizumab Drug: Nab-Paclitaxel Drug: Sacituzumab Govitecan	1-2
	NCT04427293	Preoperative Lenvatinib Plus Pembrolizumab in Early-Stage Triple-Negative Breast Cancer (TNBC)	Recruiting	Drug: Lenvatinib Drug: Pembrolizumab	1
	NCT02456857	Liposomal Doxorubicin, Bevacizumab, and Everolimus in Patients With Locally Advanced TNBC With Tumors Predicted Insensitive to Standard Chemotherapy; A Moonshot Initiative	Recruiting	Biological: Bevacizumab Drug: Everolimus Other: Laboratory Biomarker Analysis Drug: Pegylated Liposomal Doxorubicin Hydrochloride	2
	NCT03251378	A Multi-Center, Open-Label Study of Fruquintinib in Solid Tumors, Colorectal, and Breast Cancer	Recruiting	Drug: Fruquintinib (HMPL-013)	1
	NCT03797326	Efficacy and Safety of Pembrolizumab (MK-3475) Plus Lenvatinib (E7080/MK-7902) in Previously Treated Participants With Select Solid Tumors (MK-7902-005/E7080-G000-224/LEAP-005)	Recruiting	Biological: Pembrolizumab Drug: Lenvatinib	2
	NCT03720431	TTAC-0001 and Pembrolizumab Phase Ib Combination Trial in Metastatic Triple-negative Breast Cancer	Active, not recruiting	Drug: TTAC-0001 and pembrolizumab combination	1

**Table 2 cancers-13-01080-t002:** Effects of pathways dysregulation in mesenchymal breast cancer triple negative subtype.

Mechanism	Related Pathways	Effect on Tumor Cells	Effect on Immune System
EMT	TGFβ ECM receptors ALK EGFR Wnt/βcatenin mTOR PD-L1	migration metastatis chemoresistance proliferation differentiation	escape
NOTCH		proliferation differentiation MDSC accumulation Treatment-resistance	resistance
EGFR	PI3K/mTOR EMT Src	proliferation apoptosis MDSC accumulation migration angiogenesis	escape
Src	EGFR RAS PI3K/mTOR STAT	proliferation differentiation survival migration angiogenesis	escape
Angiogenesis	VEGF	growth survival	suppression

## Data Availability

Not applicable.

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
