# Peer review of "Standard of Care and Promising New Agents for the Treatment of Mesenchymal Triple-Negative Breast Cancer"

_cancers, 2021, doi:10.3390/cancers13051080_

Round 1
Reviewer 1 Report
The review entitled "Standard of care and promising new agents for the treatment of mesenchymal triple-negative breast cancer" describes an overview of the treatments for triple breast cancer. The review is well written and organized with several examples of the current treatments. In my opinion, this manuscript should be accepted for publication after minor revisions. Comments: - Abbreviations should be described when used for the first time - The authors should add a list of abbreviations - In Table 1, the caption is missing. Please add - Also, the authors should cite the Table correctly in the textAuthor Response
We wish to thank the reviewer for their positive evaluation.
The paper has been revised according to their suggestions. We covered each point as follows:
Comments:
- Abbreviations should be described when used for the first time –
abbreviations have been checked throughout the text
-The authors should add a list of abbreviations –
A list of abbreviations has been added, as suggested
-In Table 1, the caption is missing. Please add
A caption has been added [Table 1]: Comprehensive overview of ongoing clinical trials evaluating the pathways expressed in triple negative mesenchymal breast cancer as targets for therapy (page 4).
-the authors should cite the Table correctly in the text
Table 1 has been placed at the end of the period in which it was first mentioned (page 4, line 8), Subsequently, Table 1 has been cited when deemed appropriate.

Reviewer 2 Report
The aim of this review is to summarize the current knowledge with regard to the response of triple-negative breast cancer (TNBC) subtype to conventional therapeutic strategies, to promising molecular targets agents and to immunotherapy.
Although the topic is interesting and exaustively addressed by the authors, I found the manuscript quite difficolt to read being the main part a long list of clinical trials.
I believe that the manuscript can be improved by figures/cartoons briefly explaining EMT/immunosuppression and how immunotherapy can counteract immune-escape of TNBCs, in addition to simple cartoons highlighting the target of therapeutic agents within intracellular pathways. Although it might appear reduntant, since many recent reviews address this point (see Medina et al. Int. J. Environ. Res. Public Health 2020) this woud capture the interest of a broader audience.
References are mostly appropriate, however some of them are referred to other tumor types (e.g. 21, 22, 24).
Author Response
The paper has been revised according to the suggestions.
We covered each point as follows:
Reviewer 2 (round1)
English language and style are fine/minor spell check required
The style and language have been checked with the help of a native English speaker, as required; we apologize if any error has been missed.
The aim of this review is to summarize the current knowledge with regard to the response of triple-negative breast cancer (TNBC) subtype to conventional therapeutic strategies, to promising molecular targets agents and to immunotherapy.
Although the topic is interesting and exaustively addressed by the authors, I found the manuscript quite difficult to read being the main part a long list of clinical trials.
I believe that the manuscript can be improved by figures/cartoons briefly explaining EMT/immunosuppression and how immunotherapy can counteract immune-escape of TNBCs, in addition to simple cartoons highlighting the target of therapeutic agents within intracellular pathways. Although it might appear redundant, since many recent reviews address this point (see Medina et al. Int. J. Environ. Res. Public Health 2020) this would capture the interest of a broader audience.
Thanks for this observation. Certainly there are several reviews regarding triple negative breast cancer available in the literature. However, we focused our research on the particular mesenchymal subtype, assuming that in the near future clinical trials will have to be designed taking into account the different molecular subtypes of the patients. As highlighted by Medina et al., each subtype is driven by different molecular pathways and each of these is a potential target of our therapeutic options. We tried to highlight which therapies could be used in the mesenchymal one, paying particular attention to the epithelium-mesenchymal transition as an important and determinant mechanism of immunosuppression and resistance. A further table titled “Table 2: Effects of pathways dysregulation in mesenchymal breast cancer, triple negative subtype” was added, as suggested (page 10). In order to support the discussion, we added an additional illustration item summarizing the most relevant genetic alterations of the mesenchymal breast cancer, triple negative subtype and the subsequent hallmarks of cancer entitled “Figure 1: genetic alterations and hallmarks of mesenchymal breast cancer triple negative subtype”
References are mostly appropriate, however some of them are referred to other tumor types (e.g. 21, 22, 24).
In the original text it is underlined that ETM occurs in BC as well as in other cancer types. However the sentence “in BC as well as in other malignancies” has been moved to the end of the sentence as to emphasize its non-unique presence in mesenchymal tumor (page 3).

Reviewer 3 Report
This manuscript by Silvia Mezi and coworkers provides a detailed overview of approaching novel therapies for M-type triple negative breast cancer, a subtype that appears to be largely refractory to immunotherapy. The topic is well introduced an in a logical manner.
Moreover, table one provides, what appears to be a comprehensive overview of ongoing clinical trials for this disease type focusing on a variety of molecular targets that are currently being explored. These targets include the Notch pathway, PI3K/PKB/mTOR/PTEN, Wnt signaling, angiogenesis inhbitors etc. The remainder of the manuscript involves a detailed description of the different clinical trials and which of these have been succesfull and which not. It is in this part of the manuscript that the reader get lost in the rather overwhelming amount of facts being presented. I feel that the manuscript would benefit from a more organized manner in which the data are presented. First of all, the different molecular pathways could be presented in a more organized manner and in the form of different sub-chapters. Moreover, it would help the reader if a Figure was presented showing the different pathways that appear to be targetable.
With respect to describing the clinical trials try to start by pointing out whether a particular targeting approach has clinical benefits, provide a few examples that corroborate this conclusion and, where necessary, draw attention to trial results that suggest otherwise. Moreover, the authors, as they are supposed to be the experts in the field, make projections about what is likely going to be effective and what not. In other words, take the reader by the hand in showing what therapies show the most promise, rather than a rather dry description of all possible clinical trials.
-Other points: names of targeted agents and what they are targeting should be introduced
more clearly.
- Line 120 : a membrane receptors ? do the autors mean a number of membrane receptors?
- Line 136, PI3K/MTOR should be the beginning of a new paragraph with a subtitle,
- Line 142: based on these evidences better use: based on this evidence….
- Line156 : what is meant by low PTEN rate??? use PTEN protein expression
- Line 276 respresents a rather poor introduction of the prominent role of EGFR signaling in BC and deserves a more comprehensive introduction
Author Response
The paper has been revised according to the suggestions.
We covered each point as follows:
Reviewer 3 (round1) Open Review
This manuscript by Silvia Mezi and coworkers provides a detailed overview of approaching novel therapies for M-type triple negative breast cancer, a subtype that appears to be largely refractory to immunotherapy. The topic is well introduced an in a logical manner.
We wish to thank the reviewer for their positive evaluation.
Moderate English changes required
Use of English language has been checked with the help of a native English speaker, as required; we apologize if any error has been missed.
Moreover, table one provides, what appears to be a comprehensive overview of ongoing clinical trials for this disease type focusing on a variety of molecular targets are currently being explored. These targets include the Notch pathway, PI3K/PKB/mTOR/PTEN, Wnt signaling, angiogenesis inhbitors etc. The remainder of the manuscript involves a detailed description of the different clinical trials and which of these have been succesful and which not. It is in this part of the manuscript that the reader lost in the rather overwhelming amount of facts being presented. I feel that the manuscript would benefit from a more organized manner in which the data are presented. First of all, the different molecular pathways could be presented in a more organized manner and in the form of different sub-chapters. Moreover, it would help the reader if a Figure was presented showing the different pathways that appear to be targetable.
As requested, we have simplified the discussion section and structured it in a more fluid fashion. Sub-chapters have been introduced as suggested. In order to support the discussion section we added an additional illustration item summarizing the most relevant genetic alterations which can be found in mesenchymal breast cancer, triple negative subtype and the subsequent hallmarks of cancer entitled “Figure 1: genetic alterations and hallmarks of mesenchymal breast cancer triple negative subtype”
With respect to describing the clinical trials try to start by pointing out whether a particular targeting approach has clinical benefits, provide a few examples that corroborate this conclusion and, where necessary, draw attention to trial results that suggest otherwise. Moreover, the authors, as they are supposed to be the experts in the field, make projections about what is likely going to be effective and what not. In other words, take the reader by the hand in showing what therapies show the most promise, rather than a rather dry description of all possible clinical trials.
By dividing the discussion section in sub-chapters our evaluations on clinical trials results and perspectives have been highlighted more clearly in the text.
The therapies we consider particularly promising have been discussed and introduced in the text in a smoother way in order to provide the reader with the keys to understand the data, as suggested.
The central role of TGF-β has been underlined and better explained in the text (pages 3 and 4), as well as the role of EGFR. The results of the modulation of the pathway, as well as future perspectives, have now been discussed in the text (page 18).
Other points: names of targeted agents and what they are targeting should be introduced more clearly
Unfortunately, the majority of molecules tested in phase I are still unnamed, therefore they have been introduced in the table based on their mechanism of action.
Line 120: a membrane receptors? do the autors mean a number of membrane receptors?
We apologize profusely for the mistake and the misunderstanding: page 11, line 195 of the text has now been corrected in “a membrane receptor”, thank you for your feedback.
Line 136, PI3K/MTOR should be the beginning of a new paragraph with a subtitle,
As suggested the sub-chapter 3.2 entitled “PI3K/mTOR pathway” has been introduced (page 11).
Line 142: based on these evidences à better use: based on this evidence….
the mistake has been corrected in the text (line 222)
Line156: what is meant by low PTEN rate??? à use PTEN protein expression
"PTEN protein expression" has been used, as suggested (line 238)
Line 276 respresents a rather poor introduction of the prominent role of EGFR signaling in BC and deserves a more comprehensive introduction.
A comprehensive discussion of EGFR pathway has been introduced as suggested (lines 375-410) with the relative references:
“Several types of cancer, including BC, involve deregulation of EGFR-mediated sig naling caused by different molecular mechanisms, such as overexpression, acquisition of activating mutations of the receptor and activation induced by ligands, which act in autocrine/paracrine manner [78]. The receptor regulates many aspects of the tumor be havior including cell proliferation, migration, angiogenesis and is involved in develop ment and progression of BC. EGFR is frequently overexpressed in TNBC. BL2 TNBC subtype displays unique gene ontologies involving growth factors signaling. EGFR has also been implicated as a key role player in the mitogenic and motogenic effects. Recent studies have shown that EGFR regulate migration, tumor invasion and EMT. EGFR in hibitors induced a restoring from mesenchymal to epithelial phenotype in TNBC cells and the EGFR TKIs erlotinib inhibited tumor growth and metastasis in a SUM149 xenograft mouse model, showing an antimetastatic effect that could be the basis of “overlap sensitivity” to dasatinib between M and BL2 subtypes [79].”

Reviewer 4 Report
This is a very interesting, and well written, manuscript that reviews some of the treatments for the mesenchymal subtype of triple-negative breast cancer (TNBC). It provides an overview about the role of epithelial-to-mesenchymal transition on immunosuppression, and clinical trials aimed to study the suppression of some molecular pathways and angiogenesis, as well as immunotherapy in TNBC.
There are several comments that must be addressed:
- TGFβ signaling pathway is tightly correlated with acquisition of mesenchymal properties of tumor cells and is upregulated in mesenchymal TNBC. The authors indicate three clinical trials in Table 1, however, they do not describe nor discuss any single clinical trial to investigate TGFβ inhibitors. Because this signaling pathway seems to be highly relevant in mesenchymal TNBC, the authors must describe those clinical trials and discuss them as possible therapeutic options for this TNBC subtype.
- Is there any clinical trial with TGFβ inhibitors and immunotherapy? If there is, please, describe.
- It should be interesting if the authors could discuss (maybe in “Future perspectives” section) other targets involved in signaling pathways such as TGFβ, Wnt, PI3K,… that could be promising therapeutic target for mesenchymal TNBC. For example, ATF4 which is involved in TGFβ, PI3K or even Myc (Clin Cancer Res. 2018 Nov 15;24(22):5697-5709; Nat Cell Biol. 2019 Jul;21(7):889-899).
- Are there clinical trials with repurposed drugs that could be used for treating mesenchymal TNBC? Drug repurposing in TNBC has been reviewed elsewhere but the mesenchymal subtype has not addressed specifically. It would be nice if this point could be discussed.
Minor comments:
- Page 2, line 46: do they mean PR?
- Table 1 legend is missing.
- Page 2: change wnt by Wnt throughout the page.
- Page 3: change src by Src throughout the page.
- Page 6, line 419: remove “3.1” at the end of the paragraph.
- Revise the format of the references: volume and pages are missing.
Author Response
Reviewer 4 (round1)
English language and style are fine/minor spell check required
Use of English language has been checked with the help of a native English speaker, as required; we apologize if any error has been missed.
This is a very interesting, and well written, manuscript that reviews some of the treatments for the mesenchymal subtype of triple-negative breast cancer (TNBC). It provides an overview about the role of epithelial-to-mesenchymal transition on immunosuppression, and clinical trials aimed to study the suppression of some molecular pathways and angiogenesis, as well as immunotherapy in TNBC.
We wish to thank the reviewer for their positive evaluation and for their constructive comments
There are several comments that must be addressed:
1.TGFβ signaling pathway is tightly correlated with acquisition of mesenchymal properties of tumor cells and is upregulated in mesenchymal TNBC. The authors indicate three clinical trials in Table 1, however, they do not describe nor discuss any single clinical trial to investigate TGFβ inhibitors. Because this signaling pathway seems to be highly relevant in mesenchymal TNBC, the authors must describe those clinical trials and discuss them as possible therapeutic options for this TNBC subtype.
We have deepened the role of TGFβ pathways in the text as requested (lines 122-154)
2.Is there any clinical trial with TGFβ inhibitors and immunotherapy? If there is, please, describe.
The only 3 trials currently ongoing are evaluating Bintrafusp alfa (M7824), a bifunctional agent, which is able to act on both the TGF-b and PDL1 pathways pages 6 and 7.
3.It should be interesting if the authors could discuss (maybe in “Future perspectives” section) other targets involved in signaling pathways such as TGFβ, Wnt, PI3K,... that could be promising therapeutic target for mesenchymal TNBC. For example, ATF4 which is involved in TGFβ, PI3K or even Myc (Clin Cancer Res. 2018 Nov 15;24(22):5697-5709; Nat Cell Biol. 2019 Jul;21(7):889-899).
we thank the reviewer for this relevant suggestion. ATF4 has been introduced and discussed in the session Future Perspectives (lines 550-559)
“In this scenario, other promising targets such as the Activating Transcription Factor 4 (ATF4) may repre sent a valuable prognostic biomarker and therapeutic target in patients with TNBC, since
it is able to modulate TGFβ-induced aggression in TNBC via SMAD2 / 3/4 and mTORC2
signaling. ATF4 is overexpressed in TNBC patients, and in vitro studies have shown in creased levels of ATF4 in TGFβ1 treated TNBC cell lines. ATF4 is involved in the regula tion of signaling pathways associated with tumor metastases, proliferation and drug resistance. Furthermore, inhibition of ATF4 expression led to a reduction in migration,
invasiveness, proliferation, ETM, and levels of antiapoptotic and stem cell markers and
correlated with lower patient survival [102].The main objective so as to modulate the
aggressiveness of will be to identify integrated therapeutic strategies adapted to the genetic tumor identity. Patient profiling will be therefore critical in order to determine both
the first and the subsequent lines of combination treatment. Monitoring tumor signaling
molecules levels during treatment will be required in order to achieve a durable re sponse. The final goal will be a tailored combination of cytotoxic agents, target agents
and immunotherapy with the main challenge of restoring immunocompetence in M
breast cancer patients.”
4.Are there clinical trials with repurposed drugs that could be used for treating mesenchymal TNBC? Drug repurposing in TNBC has been reviewed elsewhere but the mesenchymal subtype has not addressed specifically. It would be nice if this point could be discussed.
As suggested, subgroup analysis from the phase III TNT clinical trial has been added. Unfortunately, even in this study the response of non-basal-like TNBC was not further stratified into the M and LAR subtypes.
Page 2 lines 73-85 “Gene expression analysis may also influence chemotherapy treatment choices; a randomized phase III trial was performed comparing use of carboplatin vs docetaxel in unselected advanced TNBC and in a priori specified biomarker defined sub-populations [11]. In the unselected TNBC patient population carboplatin was not more active compared to docetaxel (the standard of care). Furthermore, there was no evidence that basal-like biomarkers could be predictive of higher response to platinum. Conversely, in patients with breast cancer gene (BRCA) mutation, likely to have targetable defects in DNA repair, treatment with carboplatin doubled the response rate. Finally, the response to docetaxel was significantly better than the one to carboplatin in patients with non-basal-like TNBCs. Unfortunately, such poor platinum response results were not further stratified into M and LAR subtypes, given that the first one is known to be chemoresistant while the latter being responsive to anti androgenic hormone treatment.”
Minor comments:
1.Page 2, line 46: do they mean PR?
we mean the progesterone receptor (line 50)
2.Table 1 legend is missing.
A caption has been added. [Table 1]: Comprehensive overview of currently ongoing clinical trials evaluating the pathways expressed in triple negative mesenchymal breast cancer as targets for therapy.
3.Page 2: change wnt by Wnt throughout the page.
Wnt has been changed
4.Page 3: change src by Src throughout the page.
Src has been changed
5.Page 6, line 419: remove “3.1” at the end of the paragraph.
"3.1" has been removed
- Revise the format of the references: volume and pages are missing.
References have been revised. Volume and pages were added.
Round 2
Reviewer 3 Report
The authors have adequately addressed the comments and questions raised, which has significantly improved the readability of the manuscript.
Reviewer 4 Report
The authors have addressed all my comments.